# Effect of COVID-19 pandemic on internet gaming disorder among general population: A systematic review and meta-analysis

**Lovin Gopali**, **Rolina Dhital** *, **Rachita Koirala**, **Trijya Shrestha**,
**Sandesh Bhusal**‡, **Reshika Rimal**‡, **Carmina Shrestha**‡, **Richa Shah**‡

Health Action and Research, Kathmandu, Nepal

ʘ These authors contributed equally to this work.
‡ These authors also contributed equally to this work.
* rolina.dhital@gmail.com

**Data Availability Statement:** All relevant data are within the paper and its Supporting information files.

## Abstract

Internet gaming disorder (IGD) has been rising in recent years. The COVID-19 pandemic has led to a noticeable shift in the way people interact with technology, which could have further contributed to an increase in IGD. Post-pandemic, the concern for IGD is likely to continue as people have become increasingly reliant on online activities. Our study aimed to assess the prevalence of IGD among the general population globally during the pandemic. Relevant studies that assessed IGD during COVID-19 were identified using PubMed, EMBASE, Scopus, CINAHL, and PsycNET between 2020, Jan 1 and 2022, May 23. We used NIH Quality Assessment Tool for Observational Cohort and Cross-Sectional Studies to assess the risk of bias, and GRADEpro for the certainty of the evidence. Three separate meta-analyses were performed using Comprehensive meta-analysis software and Revman 5.4. In total, 362 studies were identified, of which 24 observational (15 cross-sectional and 9 longitudinal) studies among 83,903 population were included in the review, and 9 studies in the meta-analysis. The risk of bias assessment showed an overall fair impression among the studies. The meta-analysis for a single group of 3 studies showed the prevalence rate of 8.00% for IGD. Another meta-analysis of 4 studies for a single group showed a pooled mean of 16.57 which was lower than the cut-off value of the IGDS9-SF tool. The two-group meta-analysis of 2 studies showed no significant difference between the groups before and during COVID-19. Our study showed no clear evidence of increased IGD during COVID-19 due to limited number of comparable studies, substantial heterogeneity, and low certainty of evidence. Further well-designed studies are needed to provide stronger evidence to implement suitable interventions to address IGD worldwide. The protocol was registered and published in the International Prospective Register for Systematic Review (PROSPERO) with the registration number CRD42021282825.

**Funding:** The author(s) received no specific funding for this work.

**Competing interests:** The authors have declared that no competing interests exist.

## Introduction

In recent years, internet gaming has become a popular leisure activity with 2.96 billion gamers worldwide [1]. An internet game, often synonymized with an online game, is a video game that is either partially or primarily played through the internet or any other computer network available [2]. With its exponential growth, there is an emerging concern that internet gaming leads to behavioral addiction [3].

Earlier days of the COVID-19 pandemic led to the closure of many social and recreational physical activities, resulting in a shift towards online games. People started seeking social interaction through online gaming, such as massively multiplayer online role-playing games [4]. Moderate gaming is associated with relaxation and stress reduction, positively enhancing learning ability and increasing cognitive skills [5]. However, excessive gaming can lead to mental health issues such as depression and insomnia, lower academic achievement and psychosocial well-being, and lead to changes in eating habits, musculoskeletal problems, and have a negative impact on quality of life [6]. The addictive nature of internet gaming may also lead to internet gaming disorder (IGD).

IGD was recently recognized as a mental health disorder by World Health Organization (WHO) in 2018 and is included in the latest revision of the International Classification of Diseases (ICD-11) [7]. The American Psychiatric Association defined IGD as "persistent and recurrent use of the internet to engage in games, often with other players, leading to clinically significant distress" [8]. IGD resembles symptoms of a substance use disorder, such as loss of control, functional impairment, distress, and interference with daily activities resulting in poor physical and psychosocial health [9].

Research on IGD has been emerging before the pandemic [10]. Since the pandemic saw a significant increase in online gaming, it is essential to understand the extent of its impact on IGD, especially in the light that IGD was grouped under mental health conditions recently [7]. Systematic reviews that gave a comprehensive understanding of how the pandemic has affected IGD was limited. Therefore, this study aimed to bridge this knowledge gap and assess the prevalence of IGD among the general population during the COVID-19 period. Information obtained from this study may help identify the vulnerable groups and promote the regulatory use of internet gaming to improve the physical, mental, and psychosocial health of gamers. It will aid in raising public awareness about the disorder beyond the pandemic period since IGD is likely to persist due to increased digital reliance. This study will provide valuable insights for concerned researchers, policymakers, and program managers to develop targeted interventions to prevent and treat IGD.

## Methods

### Protocol

The manuscript was prepared according to the Preferred Reporting Items for Systematic Reviews and Meta-Analyses (S1 Table) [11]. The protocol was registered and published in the International Prospective Register for Systematic Review (PROSPERO) with the registration number CRD42021282825.

### Information source

We searched databases of PubMed, Excerpta Medica dataBASE (EMBASE), Scopus, Cumulated Index to Nursing and Allied Health Literature (CINAHL), and PsycNet from January 1, 2020, to May 23, 2022. The Boolean search strategy was used with different terms to find the exposure (online gaming, internet gaming, computer gaming, gaming behavior, digital

gaming, video gaming, online role-playing game), outcomes (abuse, addiction, disorder, effects, overuse, problematic, compulsive, excessive, pathological) and time period (COVID-19, COVID 19 pandemic, coronavirus disease). The details of the search terms used are outlined in (S2 Table).

## Eligibility criteria

Studies on IGD consisting of a global population of all ages and genders were included in this study. Studies on online internet gaming, solo or multiplayer, during the COVID-19 pandemic were included. Studies on offline games, i.e., games that do not require internet access, were excluded. Original studies estimating the prevalence of IGD during the COVID-19 period with or without a comparison group that fit the inclusion criteria were included. Studies that defined and diagnosed IGD using the definition and proposed criteria based on DSM-5 were included. Qualitative studies, case studies, viewpoints, letters to the editors, and review articles were excluded from this study. Studies with unavailable full text or missing relevant data were excluded. Only studies that were published in English were taken into consideration.

## Data extraction

The data extraction table included the following; the name of the author, article URL, decision to include or exclude the article, reason for exclusion, name of the author, year of publication, country, study design, the objective of the study, sampling, sample size, exposure, comparison group, type of analyses, internet gaming disorder values and the IGD scale. The table was adapted from the format of the data extraction table recommended by Cochrane [12]. Each study that was included after the title abstract screening was read several times by four authors (LG, RD, RK, and TS), and the information was updated accordingly on the data extraction table and a joint decision was made by the four authors regarding whether to include or exclude the study. A reason for exclusion was provided for each article that was excluded in the process.

## Data collection and analysis

Two authors (LG, RD) independently searched different databases using the search terms (S2 Table). The retrieved studies from each database were uploaded to the citation manager software, Zotero [13]. Four authors (LG, RD, RK, TS) equally divided the responsibilities to remove the duplicates. The remaining studies were screened for title and abstract to exclude irrelevant studies not fitting the inclusion criteria. While excluding the irrelevant studies, two groups were formed (TS, RD) and (RK, LG), and each group screened the studies. Full text of relevant titles and abstracts were then searched. Again, all the included studies were divided into equal halves, and the authors worked in two groups (RD, TS) and (RK, LG) while extracting the data from the full text. In case of disparities during the process, it was resolved by discussion between all the authors. The final decision was made based on the majority's decision. Study characteristics and outcomes were independently extracted from the eligible studies by all the authors.

## Primary outcomes

IGD was the primary outcome. It was measured using tools such as the Internet Gaming Disorder Scale-Short Form (IGDS9-SF), Maladaptive game use scale (MGUS) scales, Internet gaming disorder scale (developed by Cui), Internet Gaming Disorder Test (IGD-20 test), and Internet Gaming Disorder Questionnaire (IGDQ). According to the DSM-5, the diagnostic

criteria of IGD include the following clinical symptoms, five out of the nine scores are considered supportive for the diagnosis of IGD [14].

1. preoccupation with videogames ("preoccupation")

2. experiencing unpleasant symptoms when playing videogames ("withdrawal")

3. the need to spend an increased amount of time involved in video games ("tolerance")

4. failed attempts to control participation in videogames ("lose control")

5. losing interest in past hobbies and entertainment as a result of and with the exception of, videogames ("surrender from other activities")

6. continue to use videogames despite having knowledge of psychosocial problems ("continuation")

7. deceiving family members, therapists or others regarding the number of videogames ("fraud")

8. using videogames to escape or eliminate negative feelings ("escape")

9. harm or lose relationships, work, or education or significant career opportunities because of participation with videogames ("negative consequences")

### Risk of bias within studies

NIH Quality Assessment Tools for Observational Cohort and Cross-Sectional Studies was used [15]. Four reviewers in two groups (RD, TS) and (RK, LG) equally divided the studies for the quality assessment. Each reviewer worked independently, and later the results were compared. Any disagreements were resolved by further discussions among all the authors.

### Narrative synthesis

A systematic narrative synthesis of the findings from the included studies was done. The following information was extracted from each article: author name, publication year, country of study, study design, objective, population, sample size, comparison group, outcome, and type of IGD scale used. The values for mean and standard deviation scores for continuous outcomes and proportions for dichotomous outcomes were recorded for IGD diagnosis. The values for the comparison groups, if any, were also recorded.

### Meta-analysis

Datasets were analyzed using Comprehensive Meta-Analysis (CMA) for single group pooled mean and pooled prevalence and Review Manager (RevMan 5.4) for the mean difference between the two groups accordingly. Dichotomous variables and continuous variables were used whenever applicable. The random-effects model was chosen to analyze the pooled data for IGD scores. The heterogeneity of the studies was quantified by the $I^2$ statistic. $I^2$ more than 50% was considered substantial heterogeneity across studies [16].

### Certainty of evidence

Grading of Recommendation, Assessment, Development, and Evaluation (GRADE) approach was used to assess the certainty of the evidence for eligible studies. The GRADEpro tool was used for scoring and summarizing the certainty of evidence [17]. Five domains were assessed

to downgrade the scores, which included risk of bias, precision, inconsistency, indirectness, and publication bias. As the included studies were observational in design, three domains were assessed to upgrade the scores based on the magnitude of effect, opposing plausible residual bias of confounding and dose-response gradient. As all studies were observational, the initial score for the GRADE assessment was low. The scores were either downgraded from low to very low or upgraded to moderate or high quality. Depending upon the severity, the scores were devalued by one for serious concerns and two for very serious concerns.

## Results

### Study selection and characteristics

A total of 362 studies were identified from five databases using the search terms and 149 duplicates were removed. From 213 studies screened for title and abstract, only 63 reports were included for retrieval. One study could not be retrieved; hence only 62 studies were assessed for eligibility. Further, 38 studies were excluded by full-text screening, and 24 studies were included in the final review. Among the excluded studies, 28 did not have IGD as the study outcome, seven were not original research, two did not mention the tool used to measure IGD, and one did not mention COVID-19 as the exposure period. The process of screening and inclusion of the studies are outlined in the PRISMA flowchart (Fig 1).

Table 1 shows the characteristics of all the studies included in this review. Out of the total studies, two studies [18,19] were conducted globally while eight studies were from China [20–27], three studies were from Hong Kong [28–30], two studies were from India [31,32], and one study was each from South Korea [33], Vietnam [34], Italy [35], Iran [36], Nepal [37], Japan [38], and Malaysia [39]. One multi-country study included participants from New Zealand, Australia, and the USA [40], and another study included participants from Sri Lanka, Turkey, Australia, and the USA [41]. The study designs employed were cross-sectional (15/24) and longitudinal (9/24).

Out of 24 studies, 20 studies used the Internet Gaming Disorder Scale-Short Form (IGDS9-SF), while the rest of the studies used Maladaptive Game Use Scale (MGUS), Internet gaming disorder scale (developed by Cui), Internet Gaming Disorder Test (IGD-20 test), and 11-item Internet Gaming Disorder Questionnaire (IGDQ-11) to measure IGD. Twelve studies reported single group means; seven studies reported comparative means for longitudinal data, three studies reported proportions, and two studies reported median values.

The total sample size was 83,903 from all 24 studies. Eight studies included adolescents, and five studies included adults only. Three studies included only children, while the remaining included either children and adolescents or adolescents and adults.

### Risk of bias

The risk of bias assessment within the studies was performed using the NIH quality assessment tool for observational studies (Table 2). The research objective and population studied were stated clearly in all included studies. All studies except one mentioned the inclusion and exclusion criteria. However, sample size justification, power description, and effect estimate were not reported in the majority of the studies. None of the studies were blinded. Studies were categorized into poor, fair, and good on the basis of the scoring system (Table 2).

### Certainty of evidence

Table 3 shows the assessment of the certainty of evidence conducted using GRADE for three different group of studies. All the groups were graded as very low owing to the risk of bias,

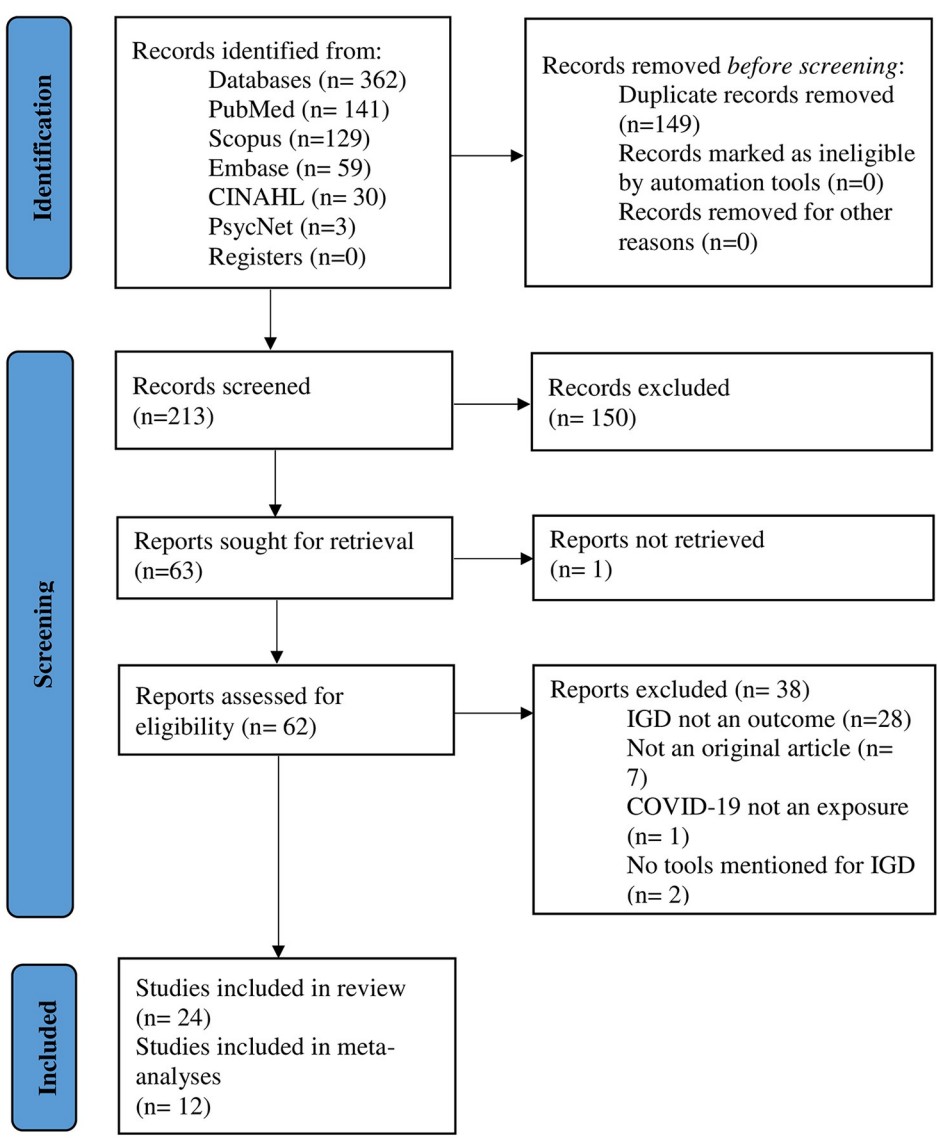

From: Page MJ, McKenzie JE, Bossuyt PM, Boutron I, Hoffmann TC, Mulrow CD, et al. The PRISMA 2020 statement: an updated guideline for reporting systematic reviews. BMJ 2021;327:n71. doi: 10.1136/bmj.n71

**Fig 1. PRISMA flowchart of the included studies.**

inconsistency, and indirectness. GRADE was not performed for single studies. There was no upgrade in the scores based on large effect size, opposing plausible residual bias, and dose gradient relationship.

**Table 1. Characteristics of the included studies.**

| SN | Study | Country | Study Design | Objective | Population | Sample size | Comparison group | Internet gaming disorder | IGD scale |
|---|---|---|---|---|---|---|---|---|---|
| 1 | Kim 2021 [33] | South Korea | Longitudinal study T1[a]: 2018 (Before COVID-19) T2[b]: 2020 (During COVID-19) | To investigate the addictive internet gaming behavior among Korean adolescents before and during the COVID-19 pandemic outbreak and examine the influence of pandemic on addictive internet gaming usage and time spent playing games on internet | Adolescents | T1-3040 T2-2906 | T1 vs T2 | Mean (SD)[c] 1) 2018: 40.69 (0.10) 2) 2020: 42.78 (0.12) | MGUS[d] |
| 2 | Wang 2022 [24] | China | Cross-sectional study | To explore whether there is a difference in the relationship of IGD with depression, anxiety, or stress, and explore the influence of fear of missing out on depression, anxiety, or stress. | Adolescents | 324 | None | Mean (SD) 1) Male: 0.22 (0.21) 2) Female: 0.14 (0.18) | Internet gaming disorder scale (developed by Cui) |
| 3 | Sallie 2021 [19] | Global (80 countries; majority from the US and UK) | Cross-sectional study | To assess the influence of COVID-19 social isolation on the online gaming (OG) and pornography viewing (PV) in the general population. | Adults | 1334- total, for OG-746 | None | Mean (SD) 1) increased weekly hours during quarantine: 12.84 (7.73) 2) decreased weekly hours during quarantine: 10.05 (8.48) 3) no change in weekly hours: 6.16 (8.20) | IGDS9-SF[e] |
| 4 | Cuong 2021 [34] | Vietnam | Cross-sectional study | To assess the prevalence of gaming disorder among Vietnamese adolescents during the first six months of the COVID-19 pandemic and to explore the association of gaming disorder with parenting practice and discipline practice. | Adolescents | 2084 | None | Percent (SE) 1) Positive: 11.60% (0.70%) 2) Negative: 88.40% (0.70%) | IGD-20 test[f] |
| 5 | Volpe 2022 [35] | Italy | Cross-sectional study | To explore the prevalence of internet addiction, excessive use of social media, problematic video gaming and binge-watching during the COVID-19 pandemic in the general population and investigate the association between social isolation, the use of digital resources and the development of their problematic use. | Adults | 1385 | None | Mean (SD) 13.10 (6.30) | IGDS9-SF |

*(Continued)*

**Table 1.** (*Continued*)

| SN | Study | Country | Study Design | Objective | Population | Sample size | Comparison group | Internet gaming disorder | IGD scale |
|---|---|---|---|---|---|---|---|---|---|
| 6 | Teng 2021 [25] | China | Longitudinal study T1: October to November 2019, before COVID-19 outbreak T2: April to May 2020, during the COVID-19 pandemic | To examine the use of videogames and IGD severity during COVID-19 and explore its association with depressive and anxiety symptoms. | Children and adolescents | 1778 | T1 vs T2 | Mean (SD) 1) T1: 1.73 (0.76) 2) T2: 1.77 (0.80) | IGDS9-SF |
| 7 | Fazeli 2020 [36] | Iran | Cross-sectional study | To examine the mediating role of psychological distress (depression, anxiety, and stress) in the association of IGD with insomnia and quality of life) among adolescents during the COVID-19 pandemic. | Adolescents | 1512 | None | Mean (SD) 19.07 (7.31) | IGDS9-SF |
| 8 | Xiang 2022 [23] | China | Longitudinal study T1: October 2020 T2: 6 months after T1 | To assess the relationship between developmental assets, self-control, and IGD among adolescents. | Adolescents | 1023 | T1 vs T2 | Mean (SD) T2: 15.51 (3.59) (Value for T1 is not provided) | IGDQ-11[g] |
| 9 | Rozgonjuk 2022 [18] | Global | Longitudinal study T1: 2019 T2: 2020 T3: 2021 | To examine the change in loneliness, family relations, and disordered gaming, during the COVID-19 pandemic, and explore the association between these variables. | AdolescentS and Adults (12–56 years) | 897 | T1 vs T2 vs T3 | Mean (SD) 1) T1: 18.29 (6.54) 2) T2: 21.92 (8.01) 3) T3: 22.80 (8.09) | IGDS9–SF |
| 10 | Hall 2021 [40] | Australia, New Zealand, US | Cross-sectional study | To examine the associations between problem gambling symptomology, excessive gaming and loot box spending among isolated and non-isolated participants. | Adults (19–80 years) | 1144 | None | Mean (SD) 1) Self-isolation/ quarantine: 7.94 (5.59) 2) Not in self-isolation/ quarantine: 8.28 (5.55) | IGDS9–SF |
| 11 | Chang 2022 [22] | China | Cross-sectional study | To explore the subtypes of IGD severity and estimate the association between these subtypes and other disorders. | 13–21 years old | 1305 | None | Mean (SD) 2.51 (2.30) | IGDS9–SF |
| 12 | Shrestha 2020 [37] | Nepal | Cross-sectional study | To assess the gaming behavior of medical college students and find out its association with stress due to COVID-19 pandemic. | Adolescents and adults | 412 college students | None | Mean (SD) 16.10 (5.47) | IGDS9-SF |

(*Continued*)

**Table 1.** (Continued)

| SN | Study | Country | Study Design | Objective | Population | Sample size | Comparison group | Internet gaming disorder | IGD scale |
|---|---|---|---|---|---|---|---|---|---|
| 13 | Wu 2022 [21] | China | Cross-sectional study | To assess the impact of COVID-19 pandemic on gaming behavior of Chinese gamers and explore the risk factors for increased gaming behavior. | Adolescent and adults (18 years old or above) | 5268 gamers | None | Median (IQR) Overall IGD: 17.00 (11.00, 23.00) 1) Non-increase group: 16.00 (10.00, 21.00) 2) increase group: 18.00 (14.00, 25.00) p-value<0.001 | IGDS9-SF |
| 14 | She 2021 [29] | Hong Kong | Cross-sectional study (School-based survey) | To measure the prevalence of probable depression and probable IGD among adolescents during the period of COVID-19 school closures and test the roles of COVID-19 stress related to schooling and online learning in depression and IGD among the adolescents. | Adolescents | 3136 | None | Mean (SD) 2.10 (2.20) | IGDS9–SF |
| 15 | Balharara 2020 [31] | India | Cross-sectional study | To investigate the gaming behavior of college students during the COVID-19 lockdown and assess its association with stress. | Adolescents and adults | 128 | None | Mean (SD) 18.00 (6.73) | IGDS9-SF |
| 16 | Chen 2021 [26] | China | Longitudinal Study Time 1: collected in mid-January, 2020 (before the COVID-19 outbreak) Time 2: collected in mid-March, 2020 (during the initial stages of the COVID-19 outbreak) Time 3: collected in early June 2020 (during the COVID-19 outbreak recovery period) | To assess the changes in problematic internet use (problematic smartphone use, problematic social media use, and problematic gaming) and changes in COVID-19-related psychological distress before the outbreak of COVID-19, during the initial stage and recovery period of COVID-19. | Children and adolescents | 504 | Group1 VS Group 2 VS Group 3 created on the basis of Latent class analysis | Mean (SD) 1) T1: 1.34 (0.55) 2) T2: 1.30 (0.50) 3) T3: 1.29 (0.53) | IGDS9-SF |
| 17 | Oka 2021 [38] | Japan | Longitudinal study T1: December 2019 (before COVID-19) T2: (July 2020 (during COVID-19) | To determine the influence of the COVID-19 pandemic on IGD and problematic internet use (PIU) and assess its risk factors. | Adults | 51246 | None | Mean (95% CI) 0.62 (0.61–0.63) (Value of SD is not provided) Overall prevalence (95% CI) 4.10% (3.90%–4.20%) | IGDS9–SF |

(*Continued*)

**Table 1.** (Continued)

| SN | Study | Country | Study Design | Objective | Population | Sample size | Comparison group | Internet gaming disorder | IGD scale |
|---|---|---|---|---|---|---|---|---|---|
| 18 | Chen 2021 [30] | Hong Kong | Longitudinal study Time 1: End of October to early November 2019 Time 2: March 2020 | To assess changes in the level of engagement in internet-related activities (smartphone use, social media use, and gaming), investigate the differences of psychological distress before and during the COVID-19 outbreak and explores the mediating roles of problematic internet-related behaviors in the causal relationships of psychological distress and time spent on internet-related activities. | Children | 535 | T1 vs T2 | Mean (SD) 1) T1: 1.42 (0.55) 2) T2: 1.32 (0.51) | IGDS9–SF |
| 19 | Singh 2021 [32] | India | Cross-sectional study | To assess and compare internet use, substance use, stress, and coping among adolescents, young adults, and middle-aged adults during the COVID-19 pandemic. | Adolescents, young adults, and middle aged-adults | 1027 Adolescent = 456 (M = 200; F = 256) Adults = 347 (M = 124; F = 223) Mid adults = 224 (M = 103; F = 121) | None | Mean (SD) 1) Adolescent: [Male: 3.40 (2.30), Female: 3.59 (2.44)] 2) Young adults: [Male: 3.28 (2.57), Female: 2.67 (2.18)] 3) Middle-aged adults: [Male: 2.35 (2.24), Female: 2.29 (2.11)] | IGDS9–SF |
| 20 | Ali 2022 [41] | Sri Lanka, Turkey, Australia, and the USA) | Cross-sectional study | To examine the psychometric properties of the IGDS9-SF in Sri Lankan university students and evaluate its measurement invariance across samples from Sri Lanka, Turkey, Australia, and the USA. | Adolescents (16–18 years) | **Sri Lanka:** 322 (73 excluded due to missing data on IGDS9-SF) **Turkey:** 244 (M = 113; F = 131) **Australia:** 738 **US:** 222 | None | Median (IQR) 1) Turkey: 13.00 (10.00–18.00) 2) Sri Lanka: 18.00 (13.00–22.00) 3) USA: 21.00 (17.00–27.00) 4) Australia: 20.00 (16.00–25.00) | IGDS9-SF |

(*Continued*)

**Table 1.** (Continued)

| SN | Study | Country | Study Design | Objective | Population | Sample size | Comparison group | Internet gaming disorder | IGD scale |
|---|---|---|---|---|---|---|---|---|---|
| 21 | Chen 2022 [27] | China | Longitudinal study Wave 1: Data were collected between October and November 2019 (a period before the COVID-19 outbreak) Wave 2: Data were collected between January 6 and 9, 2020 (the COVID-19 endemic period) Wave 3: Data were collected between March 4 and 16, 2020 (the COVID-19 pandemic period) | To assess the relationship of problematic smartphone use and problematic gaming with psychological distress among schoolchildren before and during the outbreak of COVID-19. | Children | 575 | Wave 1 vs Wave 2 vs Wave 3 | Mean (SD) 1) Wave 1: 1.48 (0.59) 2) Wave 2: 1.39 (0.56) 3) Wave 3: 1.33 (0.53) | IGDS9-SF |
| 22 | Ismail 2021 [39] | Malaysia | Cross-sectional study | To measure the prevalence of internet use and internet gaming (IG) among medical students and explore its association with anxiety during COVID-19 pandemic. | Adolescents and adults | 237 | None | Proportion 2.50% | IGDS9-SF (Malay version) |
| 23 | Yang 2021 [28] | Hong Kong | Cross-sectional study (Population-based telephone survey) | To examine the patterns and levels of social media addiction and IGD and determine its association with prolonged use of social media/internet games among the general population during COVID-19 and explore the mediation effects of psychosocial statuses on these associations. | 18 years old or above | 1070 (658 recent social media users, 177 internet gamers) | None | Proportion 9.70% | IGDS9–SF |
| 24 | Chen 2022 [20] | China | Longitudinal study Time 1: before the COVID-19 outbreak pandemic (collected from early to mid-January, 2020) Time 2: during the school closure period at the initial stages of the COVID-19 outbreak (collected in mid-March, 2020) Time 3: following the lifting of restrictions (collected in early June 2020) | To assess the psychological distress among children and evaluate the role of problematic internet gaming and perceived weight stigma. | Children | T1 = 283 T2 = 277 T3 = 272 | T1 vs T2 vs T3 | Mean (SD) 1) T1: 1.48 (0.64) 2) T2: 1.44 (0.58) 3) T3: 1.42 (0.63) | IGDS9–SF |

Time 1,b- Time 2.

c- standard deviation.

d- Maladaptive Game Use Scale (MGUS).

e- The Internet Gaming Disorder Scale-Short Form (IGDS9-SF).

f- Internet Gaming Disorder Test (IGD-20 test).

g- 11-item Internet Gaming Disorder Questionnaire (IGDQ-11).

**Table 2. Assessment of risk of bias within the studies.**

| SN | Study | Q1 | Q2 | Q3 | Q4 | Q5 | Q6 | Q7 | Q8 | Q9 | Q10 | Q11 | Q12 | Q13 | Q14 | Overall |
|---|---|---|---|---|---|---|---|---|---|---|---|---|---|---|---|---|
| 1 | Kim 2021 [33] | Y[a] | Y | Y | Y | N[b] | Y | Y | Y | Y | N | Y | N | Y | N | Fair |
| 2 | Wang 2022 [24] | Y | Y | Y | Y | N | N | Y | N/A[c] | Y | N | Y | N | Y | N | Fair |
| 3 | Sallie 2021 [19] | Y | Y | N | Y | N | N | Y | N/A | Y | N | Y | N | Y | Y | Fair |
| 4 | Cuong 2021 [34] | Y | Y | Y | Y | Y | N | N | N/A | Y | N | Y | N | Y | N | Fair |
| 5 | Volpe 2022 [35] | Y | Y | Y | Y | N | N | N | N/A | Y | Y | Y | N | Y | Y | Fair |
| 6 | Teng 2021 [25] | Y | Y | Y | Y | Y | Y | Y | N | Y | Y | Y | N | Y | N | Fair |
| 7 | Fazeli 2020 [36] | Y | Y | Y | N/A | N | N | N | N/A | Y | N | Y | N | N/A | N | Fair |
| 8 | Xiang 2022 [23] | Y | Y | N/A | Y | N | Y | Y | Y | Y | Y | Y | N | N | Y | Fair |
| 9 | Rozgonjuk 2022 [18] | Y | Y | N | Y | N | N | N | N/A | Y | N | Y | N | N/A | N | Fair |
| 10 | Hall 2021 [40] | Y | Y | Y | Y | N | N | N | N/A | Y | N | Y | N | N/A | N | Fair |
| 11 | Chang 2022 [22] | Y | Y | Y | Y | N | N | N | Y | Y | N | Y | N | N/A | Y | Fair |
| 12 | Shrestha 2020 [37] | Y | Y | Y | Y | Y | N | N | N | Y | N | Y | N | N/A | N | Fair |
| 13 | Wu 2022 [21] | Y | Y | NR | Y | N | Y | Y | Y | Y | Y | Y | NR[d] | NR | Y | Fair |
| 14 | She 2021 [29] | Y | Y | Y | Y | N | N | N | Y | Y | N/A | Y | NR | N/A | Y | Fair |
| 15 | Balhara 2020 [31] | Y | Y | NR | Y | N | N | N | Y | Y | N/A | Y | NR | N/A | Y | Fair |
| 16 | Chen 2021 [26] | Y | Y | Y | Y | N | Y | Y | Y | Y | Y | Y | NR | Y | Y | Good |
| 17 | Oka 2021 [38] | Y | Y | Y | Y | N | Y | Y | Y | Y | Y | Y | NR | N | Y | Good |
| 18 | Chen 2021 [30] | Y | Y | Y | Y | N | Y | Y | Y | Y | Y | Y | NR | Y | Y | Good |
| 19 | Singh 2021 [32] | Y | Y | Y | Y | Y | N | N | Y | Y | N/A | Y | NR | N/A | N | Fair |
| 20 | Ali 2022 [41] | Y | Y | Y | Y | N | N | N | Y | Y | N/A | Y | NR | N/A | Y | Fair |
| 21 | Chen 2022 [27] | Y | Y | Y | Y | N | Y | Y | Y | Y | Y | Y | NR | N | Y | Good |
| 22 | Ismail 2021 [39] | Y | Y | Y | Y | N | N | N | Y | Y | N/A | Y | NR | NR | Y | Fair |
| 23 | Yang 2021 [28] | Y | Y | Y | Y | N | N | N | Y | Y | N/A | Y | NR | N/A | Y | Fair |
| 24 | Chen 2022 [20] | Y | Y | NR | Y | N | Y | Y | Y | Y | Y | Y | NR | Y | Y | Good |

a- Yes.

b- No.

c- Not applicable.

d- Not reported.

## Meta-analysis

Fig 2 shows the two-group meta-analysis for the mean difference using two longitudinal studies among 2353 participants. The comparison was done between the IGDS9-SF scores before and during the COVID-19 pandemic. A random-effects model was used, and the total overall effect for the mean difference is found to be -0.05 (95% CI -0.24,0.13). There was no significant difference between the groups before and during COVID-19. The value of $I^2$ was 95.00% which represents substantial heterogeneity between the selected studies.

Fig 3 shows the single group meta-analysis for proportion using three cross-sectional studies among 2943 participants. The calculation was done using the event rate for the prevalence rate of IGD and the total sample size of each study. A random-effects model was used, and the total overall effect for the event rate is found to be 0.08 (95%CI 0.05,0.13). The value of $I^2$ was 85.80% which represents a substantial amount of heterogeneity between the selected studies.

Fig 4 shows a single-group meta-analysis of pooled means using four studies among 3667 participants using the random effects model. The pooled mean is found to be 16.57 (95% CI 13.42, 19.71), which is below the cut-off scores (36–45) of the IGDS9-SF [41] with an $I^2$ of 99.50%.

**Table 3. Assessment of certainty of evidence through GRADE.**

| of studies | Study design | Risk of bias | Inconsistency | Indirectness | Imprecision | Other considerations | Before COVID-19 | During COVID-19 | Relative (95% CI) | Absolute (95% CI) | Certainty |
|---|---|---|---|---|---|---|---|---|---|---|---|
| | | | Certainty assessment | | | | of patients | | Outcome Measures | | |
| **Comparison of IGD before and during COVID-19** | | | | | | | | | | | |
| 5 | observational studies | not serious | very serious[a] | not serious | not serious | none | 3690 | 3677 | - | MD **0.05 lower** (0.12 lower to 0.02 higher) | Very low |
| **Pooled mean scores of IGD** | | | | | | | | | | | |
| 4 | observational studies | serious[b] | not serious | serious[c] | not serious | none | 0 | 3408 | - | mean **16.32 higher** (10.10 higher to 22.54 higher) | Very low |
| **Comparision of pooled prevalence rate of IGD** | | | | | | | | | | | |
| 3 | observational studies | serious[b] | not serious | serious[c] | not serious | none | 0/0 | 0.24/2943 (0.00%) | not estimable | | Very low |

CI: confidence interval; MD: mean difference.

Author(s)

**Question:** Before COVID 19 compared to During COVID 19 in IGD among general population.

**Setting:** Two groups mean meta-analysis.

Bibliography

Explanations

a. The value of inconsistency is 0% which shows that there are no differences between the included studies.

b. The risk of bias in all included studies was graded as fair.

c. Because most of the studies are from Asia and only a small number are from other parts of the world, there is no diversity among the studies that were chosen.

## Discussion

This systematic review included 24 studies that aimed to identify the prevalence of IGD during the COVID-19 pandemic among 83,903 participants. The meta-analyses included 9 studies through the comparison of mean scores of IGD before and during COVID-19 and pooled effects of prevalence rate and means for single-group analysis. The results of the meta-analyses demonstrated no significant difference in mean scores of IGD before and during COVID-19, with single-group pooled means below the cut-off score for IGDS9-SF scales. However, a pooled prevalence of 8% IGD during COVID-19 was identified from studies conducted in three Asian countries among 2943 population [28,34,39].

This systematic review showed emerging research from different regions and economies. The representation of different countries indicates that IGD is considered an emerging mental

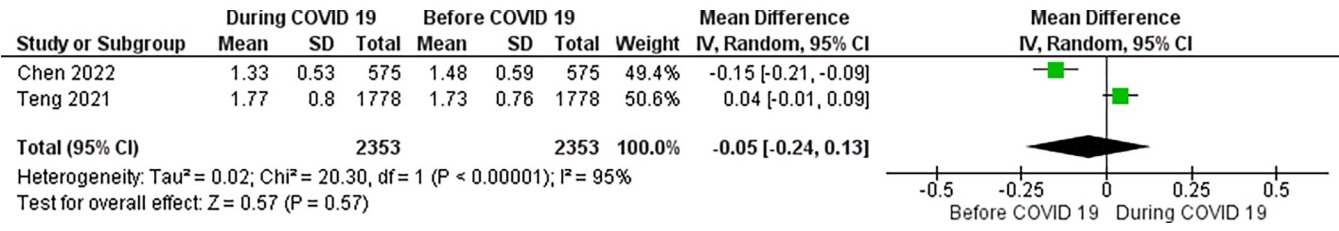

**Fig 2.**

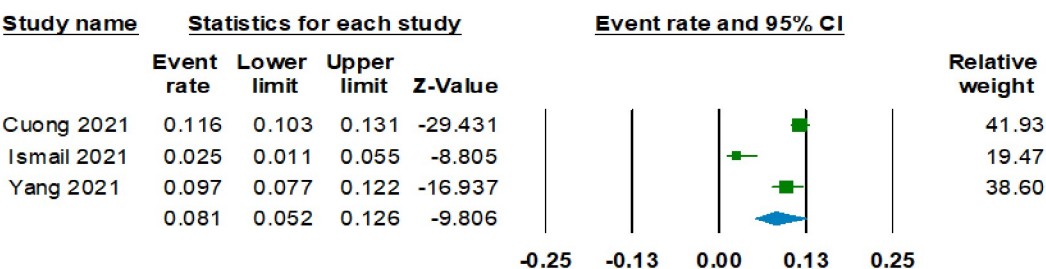

Fig 3.

health problem irrespective of the region or the economy [42]. However, the majority of the included studies in this systematic review were conducted in China. China witnessed a substantial increase in the popularity of internet games among youth and adolescents in the past two decades with 666 million online gamers in 2022, most of whom were aged more than 18 years [43]. A cross-sectional study conducted before the COVID-19 pandemic among male college students in Nanchong, China, found that one in ten students exhibited clinically significant symptoms of IGD [44]. Due to this, China has recognized IGD as a significant risk factor for mental illness and implemented numerous interventions to combat IGD and addiction, including 250 gaming detox/rehabilitation centers all over the country and restrictions on gaming hours for children under the age of 18 years [45]. Systematic and scoping reviews before COVID-19 have also shown a higher representation of data from China than in other countries [10,42].

The majority of the study population in this systematic review were adolescents. Adolescents and youth are considered to be more technology-friendly, and the use of electronic gadgets and the internet for gaming purposes is more popular among them. However, younger

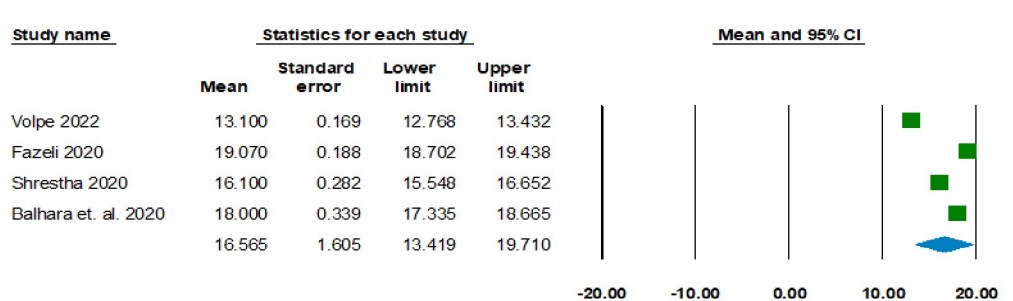

Fig 4.

people are also at risk of addictive behaviors and, thus, are considered a high-risk group for IGD [46]. They are also more prone to social isolation and lack of environmental control, which are considered significant predictors of IGD [47]. According to a systematic review from 2018, the population aged between 12 and 20 years in East Asian countries had the highest prevalence of IGD [10].

The meta-analysis of single groups showed a pooled prevalence of 8% from studies conducted in Hongkong, Vietnam, and Malaysia among 2943 adolescents and young adults [28,34,39]. The highest prevalence among the three studies was observed in the study conducted among 2084 adolescents from Vietnam, with a prevalence of 11.6% [34]. A scoping review reporting the prevalence of IGD before COVID-19 suggested a wide range of prevalence between 21% and 57.5% in the general population, with most of the studies from China, Korea, and the USA [42]. Another systematic review focusing on adolescents reported a global prevalence of 5.5% among children and adolescents before COVID-19, representing East Asia, Europe, North America, and Australia [10]. The findings of this meta-analysis indicate that the prevalence of IGD during COVID-19 may have risen in South-East Asian regions and not just limited to countries with higher economies. However, there is still a paucity of long-term data from LMICs from different regions. The findings highlight the need for more prevalence studies from different regions of the world during COVID-19.

The meta-analysis of a single group's mean IGD score was 16.3 among four studies from India, Italy, Iran and Nepal [31,35–37]. The pooled mean was lower than the mean cut-off score of 36–45 for IGD to be considered prevalent [48]. However, the studies showed substantial heterogeneity and therefore, the impression of prevalence based on pooled mean scores remains inconclusive. The two-group analysis of the mean difference based on two longitudinal studies also did not show a significant difference before and during the pandemic. There could be a lack of longitudinal studies due to a lack of baseline data before COVID-19 and feasibility issues during the pandemic resulting in underreporting of IGD.

Among the included 24 studies in this systematic review, only one study conducted in South Korea showed the highest mean for IGD before and during COVID-19, which also increased during the pandemic among high school students using MGUS (mean score of 40.69 in 2018 and 42.78 in 2020) [33]. This is consistent with the systematic review before COVID-19 that found Korea to be high, reaching an extreme 50% prevalence [10]. However, due to the use of different assessment tools for IGD, the data was not comparable with included studies from other countries. A systematic review conducted in 2018 showed that the prevalence of IGD varies widely between studies from various countries which may be due to significant differences in assessment tools, research populations, and IGD diagnostic criteria used [10].

This systematic review has certain limitations. The GRADE assessment indicated very low evidence, mostly attributed to indirectness and inconsistency. The majority of the included studies were of fair quality in the risk of bias assessment. The most common factors among these 24 studies are that most of these studies lacked a clear justification for sample size calculation and lack of association between exposure and outcome as most of the studies were cross-sectional. Thus, the findings of the systematic review highlight research gaps in this area during COVID-19 and the results could have been under-reported. We could perform meta-analysis only in 9 studies due to a lack of comparable data, and all the meta-analyses showed substantial heterogeneity. Though our studies included participants from various parts of the world, the majority of the studies were conducted in Asia, and the highest number of studies were concentrated in China. Thus, the results may not be representative of global participants. IGD was the only outcome that was measured in this study using various tools, and the results do not reflect other mental and emotional issues associated with internet gaming. Moreover,

we only included results published in English, due to which the findings could be prone to language bias. Furthermore, this systematic review was conducted amid the COVID-19 pandemic and may not have captured ongoing studies that may have the potential to yield stronger evidence.

Despite the limitations, this systematic review represented the evidence from different countries with different economies indicating emerging interest in IGD globally during the pandemic. Even though we have transitioned to normal lifestyles, the risk of IGD remains because of its addictive nature. The pandemic has changed how people interact with technology and the internet. There has been a significant increase in the use of online platforms for work, education, socialization, and entertainment during the pandemic, resulting in a rise in excessive gaming. Now that the world is adapting to the new normal after the COVID-19 pandemic, the findings of this study can be used to guide future research on IGD. Additionally, a better understanding of IGD and its prevalence during periods of stress like the COVID-19 pandemic can better help individuals who rely on the internet and online gaming for entertainment or coping with stress. The findings also highlight the research gaps globally. The findings indicate a need for better-designed studies minimizing the risk of biases and more longitudinal studies to reflect the long-term perspective of different countries and diverse groups of researchers. More robust findings on IGD during and beyond COVID-19 will provide baseline evidence to design sustainable interventions and implementation for prevention and rehabilitation programs against IGD on a global scale.

## Conclusion

This systematic review represented studies from diverse regions and economies. However, the studies were mostly from Asia, concentrating on China. Our study showed a pooled prevalence of IGD at 8% during COVID-19; however, it did not show any clear evidence of increased IGD during the pandemic. The lack of clear evidence is mainly due to limited studies with comparable data, very low certainty of the evidence of included studies, and substantial heterogeneity in the studies that were included in the meta-analysis. Nevertheless, this systematic review and meta-analyses highlight the research gap during COVID-19 for IGD, a recently recognized mental health condition. Further research on IGD is necessary to identify vulnerable groups, raise awareness, promote regulatory use of internet gaming interventions, and prevent and treat IGD beyond the pandemic period, given the likelihood of its persistence due to increased digital reliance. Better-designed studies that are comparable, reduce the risk of biases, and provide a long-term perspective should be carried out to help establish stronger evidence.

## Supporting information

**S1 Table. PRISMA statement for systematic reviews and meta-analysis.**
(DOCX)

**S2 Table. Search strategies in different search engines and registry database.**
(DOCX)

## Author Contributions

**Conceptualization:** Lovin Gopali, Rolina Dhital, Sandesh Bhusal, Richa Shah.

**Data curation:** Lovin Gopali, Rolina Dhital, Rachita Koirala.

**Formal analysis:** Lovin Gopali, Rolina Dhital, Rachita Koirala, Trijya Shrestha, Sandesh Bhusal.

**Investigation:** Lovin Gopali, Rolina Dhital, Rachita Koirala, Trijya Shrestha, Carmina Shrestha, Richa Shah.

**Methodology:** Lovin Gopali, Rolina Dhital, Rachita Koirala, Trijya Shrestha, Reshika Rimal, Carmina Shrestha, Richa Shah.

**Project administration:** Lovin Gopali, Rolina Dhital.

**Resources:** Lovin Gopali, Rolina Dhital, Rachita Koirala, Trijya Shrestha, Sandesh Bhusal, Reshika Rimal.

**Software:** Lovin Gopali, Rolina Dhital, Rachita Koirala, Trijya Shrestha, Sandesh Bhusal, Reshika Rimal, Carmina Shrestha.

**Supervision:** Rolina Dhital, Richa Shah.

**Validation:** Lovin Gopali, Rolina Dhital, Sandesh Bhusal, Reshika Rimal, Carmina Shrestha, Richa Shah.

**Visualization:** Lovin Gopali, Rolina Dhital, Rachita Koirala, Sandesh Bhusal, Reshika Rimal, Carmina Shrestha, Richa Shah.

**Writing – original draft:** Lovin Gopali, Rolina Dhital, Rachita Koirala, Trijya Shrestha, Carmina Shrestha, Richa Shah.

**Writing – review & editing:** Lovin Gopali, Rolina Dhital, Rachita Koirala, Trijya Shrestha, Sandesh Bhusal, Reshika Rimal, Carmina Shrestha, Richa Shah.

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
