## [Decision Letter · Decision Letter 0]

24 Jan 2023

PGPH-D-22-01208

Effect of COVID-19 pandemic on internet gaming disorder among general population: a systematic review and meta-analysis

Dear Dr. Dhital,

Thank you for submitting your manuscript to PLOS Global Public Health. After careful consideration, we feel that it has merit but does not fully meet PLOS Global Public Health’s publication criteria as it currently stands. Therefore, we invite you to submit a revised version of the manuscript that addresses the points raised during the review process.

We look forward to receiving your revised manuscript.

Kind regards,

Carl Abelardo T. Antonio

Academic Editor

Journal Requirements:

Additional Editor Comments (if provided):

In addition to the comments raised by the reviewers, please comment on the following concerns:

1. Some of the studies appear to be from the same author group (e.g., references 22, 29 and 31). Were authors able to assess whether these papers were reporting outcomes for different populations? If the populations are the same (i.e., same study sample), how did authors arrive at a decision to include papers in the final analysis?

2. Authors may also want to strengthen their discussion of the implications of the study in lines 343-351. In addition to mentioning what the findings mean for practitioners working with IGD-affected individuals and policymakers who might regulate internet gaming in a pandemic context, it might also be important to mention how readers should situate the findings in the now normal period.

Reviewers' comments:

Reviewer's Responses to Questions

**Comments to the Author**

1. Does this manuscript meet PLOS Global Public Health’s publication criteria? Is the manuscript technically sound, and do the data support the conclusions? The manuscript must describe methodologically and ethically rigorous research with conclusions that are appropriately drawn based on the data presented.

Reviewer #1: Yes

Reviewer #2: Partly

2. Has the statistical analysis been performed appropriately and rigorously?

Reviewer #1: No

Reviewer #2: Yes

3. Have the authors made all data underlying the findings in their manuscript fully available (please refer to the Data Availability Statement at the start of the manuscript PDF file)?

Reviewer #1: Yes

Reviewer #2: Yes

4. Is the manuscript presented in an intelligible fashion and written in standard English?

Reviewer #1: Yes

Reviewer #2: Yes

5. Review Comments to the Author

Reviewer #1: Effect of COVID-19 pandemic on internet gaming disorder among general population: a

systematic review and meta-analysis

Abstract:

1. “home confinement during COVID-19 may have contributed to this to some extent”, please rephrase this sentence in accordance with your findings, as the matter is past now and you might have the recorded effects in this regard.

2. “Our study aims to assess”, please use past tense everywhere.

3. The reader does not get a clear-cut concluding statement out of abstract, what came out of this study? Did the pandemic effect IGD and how much?

Introduction:

1. Why were the researchers interested only in the IGD? Please provide a solid rationale of this study.

2. The last paragraph of Introduction seems as if it is written in a hurry. Please decorate it further with a justified significance and scope of this study.

Results:

1. Line spacing within the tables can be reduced to have a better symmetry.

2. Table 3 seems to be the main finding of the study. However, it does not signify the statistically proven effect of the pandemic on IGD. How the authors would determine the “effect” without a relevant statistical procedure such as regression, t-test, or ANOVA.

Conclusion:

1. “Therefore the evidence on the prevalence of IGD during COVID-19 remains inconclusive”, having said that the authors must focus more on the need and rationale of the study, what made them think that IGD might have been increased during the pandemic and why the results came otherwise?

2. “Nevertheless, this systematic review and meta-analyses highlight the research gap for emerging mental health problems such as IGD during the pandemic”, please reconsider this statement as thousands of mental health related studies have already been carried out during the pandemic.

3. Authors should write some feasible suggestions based on their findings. For instance, they can focus more on the addictive properties of IGD that are quite effective even in the non-pandemic normal situations and have the capacity to disturb the daily routines.

Reviewer #2: I commend the authors for writing such an interesting article. The manuscript paper does have a few flaws, and the text might benefit from more thorough editing. I've reviewed it carefully and have some improvements to suggest. Please address the following comments and recommendations to strengthen your manuscript.

1. Please clarify if the data extraction sheet was pilot tested before actual use. If yes, please note this in your manuscript. Also, provide more details about literature summary tables and how they were developed and tailored to meet the review needs

2. Your rationale/ justification section needs to address the research gap in the intended field so that it reflects why this study is important to conduct now.

3. The novelty of your work must be discussed, it seems it is weak

4. In the last paragraph of the introduction, you should fully explain why you used systematic review and meta-analysis

5. There are some spelling and grammatical errors in the text. Please correct them

6. In the result and discussion section, more interpretations are needed -Emphasize the new research findings and mention what your research added to the previous research

6. PLOS authors have the option to publish the peer review history of their article (what does this mean?). If published, this will include your full peer review and any attached files.

**Do you want your identity to be public for this peer review?** For information about this choice, including consent withdrawal, please see our Privacy Policy.

Reviewer #1: No

Reviewer #2: No

---

## [Editor Report · Decision Letter 1]

13 Mar 2023

Effect of COVID-19 pandemic on internet gaming disorder among general population: a systematic review and meta-analysis

PGPH-D-22-01208R1

Dear Dr. Dhital,

We are pleased to inform you that your manuscript 'Effect of COVID-19 pandemic on internet gaming disorder among general population: a systematic review and meta-analysis' has been provisionally accepted for publication in PLOS Global Public Health.

Best regards,

Carl Abelardo T. Antonio

Academic Editor
